# Attention-Oriented Deep Multi-Task Hash Learning

Letian Wang [1,†], Ziyu Meng [1,†], Fei Dong [2], Xiao Yang [1], Xiaoming Xi [1] and Xiushan Nie [1,*]

1   School of Computer Science and Technology, Shandong Jianzhu University, Jinan 250101, China
2   School of Journalism and Communication, Shandong Normal University, Jinan 250358, China
*   Correspondence: niexiushan19@sdjzu.edu.cn
†   These authors contributed equally to this work.

**Abstract:** Hashing has wide applications in image retrieval at large scales due to being an efficient approach to approximate nearest neighbor calculation. It can squeeze complex high-dimensional arrays via binarization while maintaining the semantic properties of the original samples. Currently, most existing hashing methods always predetermine the stable length of hash code before training the model. It is inevitable for these methods to increase the computing time, as the code length converts, caused by the task requirements changing. A single hash code fails to reflect the semantic relevance. Toward solving these issues, we put forward an attention-oriented deep multi-task hash learning (ADMTH) method, in which multiple hash codes of varying length can be simultaneously learned. Compared with the existing methods, ADMTH is one of the first attempts to apply multi-task learning theory to the deep hashing framework to generate and explore multi-length hash codes. Meanwhile, it embeds the attention mechanism in the backbone network to further extract discriminative information. We utilize two common available large-scale datasets, proving its effectiveness. The proposed method substantially improves retrieval efficiency and assures the image characterizing quality.

**Keywords:** deep hashing; attention; multi-task learning; deep learning; image retrieval





## 1. Introduction

The approximate nearest neighbor search (ANNS) [1] serves as a crucial algorithm applied in many computer fields, such as large image retrieval, computer vision, data mining, etc. Actually, hashing is a typical ANNS algorithm. Depending on its low storage, high performance and other advantages, various retrieval tasks can be efficiently performed by it. Specifically, there are two categories of hash methods: data-independent methods [2] and data-dependent methods [3]. Data-independent methods often generate hash functions at random or artificially. For example, locality sensitive hashing (LSH) [4] adopts random projection to generate hash codes of raw data, which mostly rely on relatively long hash codes to obtain satisfactory accuracy. Alternatively, the data-dependent methods use the original data to learn a hash function that maps the data into low-dimensional binary codes to obtain a succinct representation.

According to the utilization of data labels, data-dependent hashing methods can be divided into supervised hashing methods and unsupervised hashing methods. Unsupervised methods include spectral hashing (SH) [5], iterative quantization (ITQ) [6], multi-matrix factorization hashing (MFH) [7], etc. Specifically, unsupervised methods exclusively learn hash functions and hash codes from the association among training data [8]. Compared to data-independent hashing methods, the unsupervised data-dependent hashing methods improve the retrieval performance. However, they ignore the label supervision, causing lower performance. As opposed to unsupervised hashing methods, supervised hashing methods use supervised information to increase retrieval accuracy. Early supervised hash learning methods are mainly non-deep supervised methods, including supervised

discrete hashing (SDH) [9], column sampling based discrete supervised hashing (COS-DISH) [10], fast supervised discrete hashing (FSDH) [11], scalable supervised discrete hashing (SSDH) [12], supervised discrete hashing for cross-linear regression (SDHMLR) [13], and fast scalable supervised hashing (FSSH) [14].

Recently, the deep hashing method [15–20] has become a research hotspot, such as pairwise labels-based supervised deep hashing (DPSH) [15], deep Cauchy hashing (DCH) [16], deep hashing network (DHN) [17], deep discrete supervised hashing (DDSH) [18], triplet labels based deep supervised hashing (DTSH) [19], deep semantic sorting based hashing (DSRH) [20], etc. In particular, pairwise label based supervised deep hashing (DPSH) [15] introduces a pretrained CNN-F to learn hash codes end to end in a deep network using the pairwise label similarity matrix as supervision information. Deep Cauchy hashing (DCH) [16] introduces the Alexnet network and adopts the Cauchy algorithm to optimize hash codes. Deep hashing network (DHN) [17] employs the stacking of multiple pooling layers to generate binary codes. It designs corresponding cross-entropy layers to learn similarity, and applies quantization items to reduce information loss. Triplet label-based deep supervised hashing (DTSH) [19] adopts a parameter-sharing three-branch deep neural network and combines the similarity of triple labels to obtain the binary codes with complex semantics. Deep semantic ranking based hashing (DSRH) [20] also uses deep networks to learn image representations, and further combines a ranking table encoded with multiple semantic information to learn hash codes. These methods all use nonlinear deep architecture for feature learning, and consequently, their model accuracy is significantly improved than non-deep methods.

Similarly, the attention mechanism has widely been studied as core content in deep learning. The attention mechanism [21] was first proposed by describing the global dependencies of input data and applying them to machine translation. The compression and excitation network (SENET) [22] is the first network to implement attention learning by weighting feature maps from the channel level. The convolutional block attention module (CBAM) [23] utilizes convolutional attention modules to enrich attention maps from channel and spatial dimensions. Recently, some deep hashing methods gradually incorporated the idea of the attention mechanism. For instance, the attention-aware deep adversarial hashing (AADAH) [24] pays attention to regional discriminative information and word segments that generate binary codes based on attention and adversarial mechanisms. Object location aware hashing (OLAH) [25] learns masks based on the core of attention mechanism to extract important image regions and generate corresponding hash codes. Particularly, it [26] introduces a multi-head attention unit to generate impact factors for each step output by LSTM, while paying attention to various conditions of spoken segments to generate hash codes for speech search tasks. Numerous studies have shown that it is effective to apply the attention mechanism to hash learning.

However, the majority of current hashing methods need to predefine the fixed length of the hash code. Subsequently, the model needs to be retrained again, causing the significant consumption of computing resources. Essentially, a hash code is a compact representation of the original sample, and various lengths of hash codes can represent various samples, which means that these multiple lengths binary codes will reflect different degrees of semantic information of samples. It is just that the semantic richness of them is different, but they still reveal overall semantic descriptions of the image. Possibly, there should be a certain potential correlation among them. Using these potential complementarities and correlations may strengthen the hash code quality. In our early work on supervised discrete multiple-length hashing [27], we also showed that multi-length hash codes can be considered correlated features. The goal of attention-oriented deep multi-task hash learning (ADMTH) is to generate multi-lengths hash codes concurrently. Furthermore, their contained semantic relevance can be fully utilized. Specifically, based on multi-task learning theory, supposing that $V^{(i)}$ and $V^{(j)}$ are different conceptions of a sample, their

possibility of expressing the original data error is denoted by $P_{false}(\cdot)$ , and the hypothetical potential association between them is shown in Equation (1):

$$P(V^{(i)} \neq V^{(j)}) \geq \max\{P_{false}(V^{(i)}), P_{false}(V^{(j)})\}. \qquad (1)$$

It can be seen from the formula that the upper limit of the possibility of inconsistency among two conceptions is the error possibility of any conception, so mining the association of diverse conceptions can reduce the error rate of each conception representation. The above-mentioned multi-perspective theory has been extensively studied in some works [28–30].

In a nutshell, this study devises a deep hashing framework for image retrieval, namely, ADMTH. From the perspective of network architecture, it integrates a deep convolutional neural network, a channel attention module, and a sub-network with multiple branches sharing hard parameters into a joint framework that can concurrently learn various length hash codes. This paper contributes threefold, mainly:

- To further extract discriminative fine-grained features, a channel attention module (CAM) is introduced immediately after the convolutional neural network. The important features are weighted at the channel level to enhance the expressive capability of deep features.
- To solve some drawbacks of single hash code learning, the paper designs a sub-network with multiple branches sharing hard parameters. The outputs of these branches correspond to hash codes of various lengths for a image, which greatly decreases the overhead of model computation.
- To explore the potential semantics involved in multiple hash codes, the paper designs a consistency loss based on adjacent branch hash codes, causing the smaller semantic gap among adjacent branches, which contributes to the representation capability of binary codes.

By the way, the proposed method is an upgrade version of former work [31] published in the 2021 ACM Multimedia Asia conference, which differs the conference paper in two major ways: (1) The ADMTH adds a channel attention module (CAM) in the feature extraction module to promote the expressive capability of deep features. (2) More extensive experiments are conducted in this work. Specifically, we not only conduct new comparison and ablation experiments but also conduct training efficiency comparison and time complexity analyses of ADMTH.

Research Statement: This study content is established on multi-task learning and hashing theories, which utilizes the attention mechanism and deep neural network, coming up with a deep supervised hashing method. Its purpose is enabling the deep model to learn multi-length hash codes simultaneously, reducing the time cost and computing resources caused by the repeated training of most existing single-task hashing methods. Meanwhile, this study further explores and analyzes the potential semantic associations existing among multiple hash codes, designing an objective function to optimize generated multiple hash codes: pairwise similarity and consistency loss. While ensuring the training efficiency, it also improves the characterization of query images.

The remainder of the article roughly includes four parts. Section 2 shows the technical routes of ADMTH comprehensively. Section 3 describes the experimental results comprehensively. Section 4 discusses the related theories generally. Section 5 makes a summary and gives prospects of the full paper briefly.

## 2. Proposed Method

In next subsections, we present notations and problem definitions firstly and further elaborate the ADMTH in detail, including the architecture and objective function.

### 2.1. Notations

In this paper, we apply bold lowercase letters (such as $\mathbf{v}$) to denote vectors and bold uppercase letters (such as $\mathbf{V}$) to denote matrices. The symbol $\mathbf{V}^T$ is the transpose of the matrix; the symbol $\| \cdot \|$ is the F-Norm; and the rest of the variables are italicized. The $\tanh(\cdot)$ function can constrain the values of vector elements between $-1$ and $1$. The pairwise label is provided as supervision information. Assuming that there are $N$ sample points in the database, the pairwise similarity matrix based on label information among samples is defined as $\mathbf{S}=\{s_{ij}\}^{N \times N}$. For a dataset described by multiple labels, an image contains at least multiple class labels, and the class label form is given in the form of one-hot encoding. Specifically, if two samples are similar (the two images contain at least one same class label), $s_{ij} = 1$; otherwise, (the two images do not have a common class label) $s_{ij} = 0$. Pairwise labels here usually refer to human-provided semantic labels.

### 2.2. Problem Definitions

Additionally, supposing that the task goal is to obtain $K$ hash codes, defining their corresponding length is $L_k$ ($1 < k \leq K$). The hash matrix of all $N$ sample points for the $k^{th}$ hash code can be defined as $\mathbf{H}_k = \{\mathbf{b}_i^k\}_{i=1}^{N}$, where $\mathbf{b}_i^k$ is the $k^{th}$ hash code of the $i^{th}$ sample point under the length $L_k$.

### 2.3. Architecture Structure of ADMTH

Figure 1 shows the overall framework of ADMTH. It consists of three modules: Feature Extraction Module, Channel Attention Module and Hash Learning Module. All modules are embedded end-to-end into a joint framework, concurrently learning multiple hash codes.

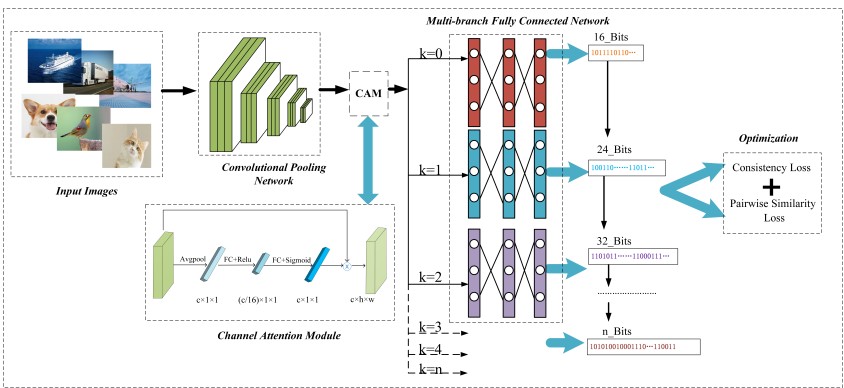

**Figure 1.** The overview architecture of ADMTH.

#### 2.3.1. Feature Extraction Module

The feature extraction module of the ADMTH is mainly constructed by fusing the typical convolutional neural network (CNN) structure and attention mechanism, and a sub-branch network in accordance with the concept of multi-task learning [32–36]. Generally, there are two types of multi-task learning: hard parameter sharing [37] and soft parameter sharing [38]. The former method means that each task guarantees the consistency of the bottom parameters learning, meanwhile, distinguishing top features learning by multiple independent parameters. In contrast to the above, the latter method means the independence of global feature learning. In other words, each task has its own set of parameters and basically no parameter sharing among them. It can be found that the hard parameter sharing method has fewer parameters, which contributes to efficiency. Specifically, in ADMTH, the learning of hash codes of different lengths can be viewed as multiple objective tasks. The part of parameter sharing occurs in the front five layers of the feature extraction network. After that, each sub-network parameter is not shared but iteratively converted in conformity with the corresponding target hash code. Based on

differentiated learning, the model can generate the binary code representation to the same image with different lengths very well.

### 2.3.2. Channel Attention Module

In addition, the ADMTH combines a streamlined SE-Net [22] immediately after the deep convolutional pooling network named the channel attention module. It focuses on weighting the important regional features of the image from the channel level, weakening the unimportant features, adaptively adjusting the feature response value of each channel, and modeling the internal dependencies between channels. It enriches the image features' output by the deep convolutional pooling network. Firstly, the depth features are compressed along the channel dimension by the average pooling operation, which reflects the non-local semantics of the image, as shown in the following Equation (2):

$$\mathbf{G}_c = F(\mathbf{D}_c) = \frac{1}{W \times H} \sum_{i=1}^{W} \sum_{j=1}^{H} \mathbf{D}_c(i, j), \tag{2}$$

where the symbol $\mathbf{D}_c$ represents the output features through the convolutional pooling network, the symbol $\mathbf{G} \in \mathbf{R}^{C \times H \times W}$ represents the compressed features, the symbol $\mathbf{G}_c$ represents the features of a certain channel dimension, and the symbol $F$ stands for channel compression operation.

Next, after the first fully connected layer, the feature dimension is reduced to $1/16$ of the input, and a ReLU is used to obtain non-linear features, which better fits the complex features of different channels. Then, the activated features go through the second fully connected layer, which aims to integrate each channel significance, restoring the dimension to the original feature dimension, and produce a weighting factor for each channel as follows:

$$\mathbf{E}_c = \sigma(W_2 \rho(W_1 \mathbf{G})), \tag{3}$$

where the symbol $\mathbf{E}_c$ represents the features after two fully connected layers, the symbol $\mathbf{G}$ represents the compressed features. The parameter $W_1$ and $W_2$ represents the weight of the two fully connected layers, respectively. The $\rho$ and $\sigma$ represents the ReLU activation function and sigmoid activation function, respectively.

Finally, we weighted the original features in the channel dimension. The $\mathbf{f}_c$ is used as an intermediate result, considered the input of the multi-branch network as shown in the following Equation (4):

$$\mathbf{f}_c = \mathbf{D}_c \cdot \mathbf{E}_c. \tag{4}$$

### 2.3.3. Hash Learning Module

Theoretically, the continuous binary code is taken by quantizing the model output. However, to avoid the complexity of optimizing the quantization loss, the element values of outputs are often restricted to the real-valued continuous space by $\tanh(x)$ in a new hash layer. In this way, it resolves the Np-Hard issue causing by optimizing the 0 and 1 directly, which are commonly used in existing deep hashing methods. In most instances, the model's optimization is actually founded on a non-binary hash code. When the model is tested, the real binary code is obtained through the symbolic function *sgn*. The kernel component of the model is the optimization of the hash code, which mainly includes two components: similarity-based loss and consistency-based loss. Making generated binary codes better reflect the semantic association with the original sample is achieved by minimizing the two loss items.Next, the above optimization is illustrate exhaustively in the Objective Function section.

### 2.4. Objective Function

Preserving similarity is an essential property of hash codes [9–13]. Therefore, the proposed method utilizes pairwise similarity matrix as supervision information to construct

the similarity loss $P_{Loss}$. This loss term is to establish the relevance between Hamming space and original sample label space so that the data distribution of the Hamming space is closer to the distribution of the original sample label space. More specifically, the larger the similarity samples, the smaller the metric distance among the corresponding binary codes. On the contrary, the lower the similarity samples, the larger the metric distance between each other. Theoretically, $P_{Loss}$ adopts the maximum a posteriori estimation of the hash code matrix **H** in accordance with the pairwise similarity label matrix **S** to construct the loss function. As shown in Equation (5),

$$P_{Loss}(\mathbf{S}, \mathbf{H}) = p(\mathbf{H}|\mathbf{S}) \propto p(\mathbf{S}|\mathbf{H})p(\mathbf{H}) = \prod_{s_{ij} \in \mathbf{S}} p(s_{ij}|\mathbf{H})p(\mathbf{H}), \tag{5}$$

where the $p(\mathbf{S}|\mathbf{H})$ represents the similarity function, the $p(\mathbf{H})$ represents prior distribution. Moreover, the $p(s_{ij}|\mathbf{H})$ reveals the resemblance between image pairs based on given images hash matrix **H**. Then, we have

$$p(s_{ij}|\mathbf{H}) = \begin{cases} \xi(\phi_{ij}), & s_{ij} = 1 \\ 1 - \xi(\phi_{ij}), & s_{ij} = 0 \end{cases}, \tag{6}$$

where the $\xi(x) = 1/(1 + e^{-x})$ means the sigmoid function, and the $(\phi_{ij}) = \frac{1}{2} < \mathbf{b}_i, \mathbf{b}_j >= \frac{1}{2}\mathbf{b}_i^T\mathbf{b}_j$. Furthermore, Equation (7) can be obtained as follows:

$$\begin{aligned} P_{Loss}(\mathbf{S}, \mathbf{H}) &= -\sum_{s_{ij} \in \mathbf{S}} \log p(s_{ij}|\mathbf{H}) \\ &= -\sum_{s_{ij} \in \mathbf{S}} s_{ij}\mathbf{b}_i\mathbf{b}_j^T - \log(1 + e^{\mathbf{b}_i\mathbf{b}_j^T}). \end{aligned} \tag{7}$$

Among them, symbol **H** represents all samples' binary matrix, symbol **S** denotes the pairwise similarity matrix, and $\mathbf{b}_i$ and $\mathbf{b}_j$ represent the hash codes of the $i^{th}$ and $j^{th}$ sample, respectively.

In addition, to further optimize the multiple hash codes of a sample, this paper designs a consistency loss in accordance with the output of adjacent branches. The semantic gap among these multiple hash codes is narrowed by the proximity of adjacent branches. Based on the previous problem definition, the hash code matrix of all sample points of length $L_k$ and $L_{k+1}$ under the $k^{th}$ and $(k+1)^{th}$ branch is obtained, which can be defined as $\mathbf{H}_k$ and $\mathbf{H}_{k+1}$, respectively. The symbol **Q** represents the learned mapping matrix, mapping from $\mathbf{H}_{k+1}$ to $\mathbf{H}_k$. The specific $C_{Loss}$ is shown in Equation (8):

$$C_{Loss}(\mathbf{H}_k, \mathbf{Q}_k^T\mathbf{H}_{k+1}) = \sum_{k=1}^{K} \| \mathbf{H}_{k+1}\mathbf{Q}_k^T - \mathbf{H}_k \|_2 . \tag{8}$$

Finally, by combining Equation (5) to Equation (8), the total loss of ADMTH is written as Equation (9). The $\alpha$ and $\beta$ are the net-parameters.

$$\begin{aligned} \min_{\mathbf{b}_i, \mathbf{b}_j, \mathbf{Q}_k^T, \mathbf{H}_k, \mathbf{H}_{k+1}} \mathcal{L} = &-\sum_{s_{ij} \in \mathbf{S}} \alpha s_{ij}\mathbf{b}_i\mathbf{b}_j^T - \log(1 + e^{\mathbf{b}_i\mathbf{b}_j^T}) \\ &+ \sum_{k=1}^{K} \beta \| \mathbf{H}_{k+1}\mathbf{Q}_k^T - \mathbf{H}_k \|_1 \\ s.t. \quad &\mathbf{H}_k \in \{-1, 1\}^{L_K \times N}. \end{aligned} \tag{9}$$

## 3. Experiments

To prove the superiority of ADMTH, we deliberate the evaluation of this method on two benchmark datasets, NUS-WIDE [39] and MS-COCO [40].

### 3.1. Datasets

Both NUS-WIDE and MS-COCO are large-scale image datasets annotated by professionals. These images are defined by multiple semantic tags and generally collected from the Internet, including landscapes, people, activities etc. Practically for the NUS-WIDE dataset, when used in image retrieval tasks, it takes into account the incomplete labeling of some classes, missing images and other disadvantages. In basically all research on hashing, the common 21 classes of image subsets are selected by default. As for the MS-COCO, we typically use the common 81 categories for image retrieval. The specific datasets information and division are shown in Table 1 below. For the fairness of the experiment, the comparison methods also adopt the same method.

**Table 1.** Datasets composition and partition way.

| Dataset | Details | | | |
|---|---|---|---|---|
| | Labels | Total Numbers | Training Set | Test Set |
| NUS-WIDE | Multiple (21) | 195 K | 10.5 k | 2.1 K |
| MS-COCO | Multiple (81) | 122 K | 10 k | 5 K |

### 3.2. Experimental Settings

Firstly, all input images sizes are generally adjusted to 224 × 224 pixels. The proposed method is initialized by using a pretrained CNN-F model. Other deep hashing methods [15–20] also adopt similar initialization strategies. As for the channel attention mechanism, we adopt empirical settings and multiple cross-experiment verifications, and the number of c (channel) is set as 16 to perform channel-level operations on the feature map. Other pivotal settings are listed as follows: The programming implementation schema adopts Pytorch, meanwhile, the total loss is iterated with built-in SGD optimizer. The selected batch size is set as 128, and the periodic learning rate decay is adopted. The initial learning rate is set as 0.15, and during the iterative process, it is set to be 1% of the previous one every 30 iterations. The symbols $\alpha$ and $\beta$ are the parameters set as 20 and 1, respectively. Specifically, we will study in the parameter sensitivity experiment later. Furthermore, we use a common metric in retrieval called mean average precision ($MAP$) to evaluate the experimental performance, which is defined as follows:

$$MAP = \frac{1}{N} \sum_{n=1}^{N} AP(i), \tag{10}$$

where the symbol $N$ represents the number of samples to be retrieved. $AP(i)$ denotes the average precision of the $i^{th}$ retrieved sample.

### 3.3. Ablation Study

As shown in Figure 2, the single hash code learning network (a) can be further subdivided to part (b) and part (c) sub-networks, based on the parameter sharing theory of multi-task learning. The turning point of the parameter sharing pattern can be in the different convolutional layer or the fully connected layer, causing various multi-task learning styles. When other experimental settings are the same, in order to simplify the training, we remove the CAM and choose the best model results in both categories for comparison. Taking the dataset NUS-WIDE as an example, DMTH-Conv and DMTH-Fullcnt represent the optimal model split at the convolutional and fully connected layers, respectively, taking the retrieval accuracy on the collection of hash codes {16, 32, 48, 64} after the model convergence.

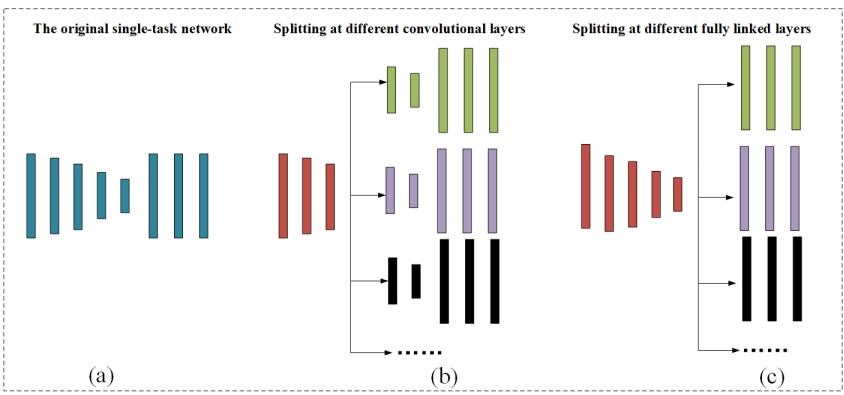

**Figure 2.** Different ways of parameters sharing. The subfigure (**a**–**c**) represents three types of structures respectively: single-task networks, networks split at the convolutional layers, and networks split at the fully connected layers.

In Table 2, the bolded value indicates the optimal performance, and observations demonstrate that the optimal of splitting at the fully connected layer is enormously better than that at the convolutional layer. One possible reason is that the shared learning of more detailed shallow features contributes to learning the last high-level representations. Therefore, we choose to split the architecture at the first fully connected layer to form a multi-branch structure.

**Table 2.** Multi-task learning split validation experiments on NUS-WIDE.

| Split Method | NUS-WIDE | | | |
|---|---|---|---|---|
| | 16 bits | 32 bits | 48 bits | 64 bits |
| DMTH-Conv | 0.610 | 0.684 | 0.725 | 0.768 |
| DMTH-Fullcnt | **0.740** | **0.771** | **0.793** | **0.801** |

Multi-task learning and attention module are the core components of the ADMTH. To verify its effectiveness, we conduct multi-task learning and attention module ablation experiments. It should be noted that to simplify training for the effective experiment of multi-task learning, we only compare the multi-branch network (with the attention module removed) and the single-branch network (with the multi-branch network and consistency loss and attention mechanism removed), respectively, taking the retrieval accuracy on the collection of hash codes {16, 32, 48, 64} after the model convergence similarly. Both experiments are performed on NUS-WIDE dataset.

Figure 3 shows a comparative experiment of simultaneous generation of multiple lengths hash codes via multi-task and decentralized generation of a set of corresponding hash codes via single-task, respectively. The results indicate that simultaneously learning multiple length hash codes with the optimization of $P_{Loss}$ and $C_{Loss}$ could better excavate the potential semantic relationship among these, further enhancing the image representation capability. Figure 4 demonstrates the significance of introducing the channel attention module, which manifests that the learning of shallow features is directly related to that of the final high-level representation, making the module better emphasize fine-grained information among image categories.

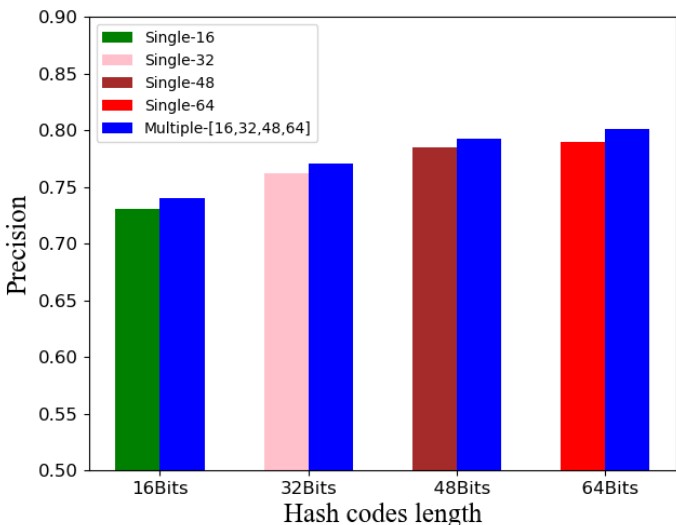

**Figure 3.** Multi-task learning ablation experiment on NUS-WIDE.

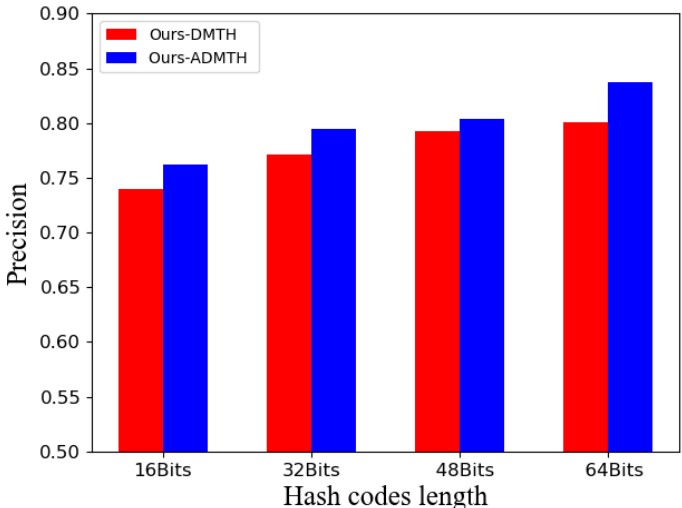

**Figure 4.** Attention mechanism ablation experiment on NUS-WIDE.

### 3.4. Time Complexity and Training Efficiency

We also introduce the time complexity analysis and training efficiency comparison experiments to certify the time efficiency of multi-task learning. We use $TC(\cdot)$ to reveal the time complexity of the method. Accordingly, given a maximum length $L_{max}$ of the hash code, the $TC(\text{ADMTH})$ for obtaining it is $O(ncL_{max}^2)$, where $c$ and $n$ are the number of labels and samples, respectively. The $TC(\text{ADMTH})$ for learning the projection $\mathbf{Q}$ is $O(ndL_{max} + nd^2)$, where $d$ denotes the input dimension. In summary, the final $TC(\text{ADMTH})$ for training multi-hash codes is $O(ncL_{max}^2 + nd^2)$.

Furthermore, based on the above-mentioned multi-task ablation experiments, we respectively record the time of training 16 bits, 32 bits, 48 bits, and 64 bits hash codes for single-task learning. Correspondingly, recording the time of multi-task trains a set of hash codes (16 bit, 32 bits, 48 bits, and 64 bits). It is worth noting that both of them converge within 150 epochs. For the fairness of comparison, we use 150 epochs as a node and record the training time every 10 epochs, calculating the average training time within 150 epochs. As shown in Table 3, the bolded value indicates the optimal performance, and the average duration of multi-task learning to train multiple lengths binary codes is more time saving than the total average duration of single-task learning. Hence, the time complexity analysis

of multi-task learning and the comparison of training efficiency experiments show the efficiency of multi-task learning, which greatly moderates computing power and time costs.

**Table 3.** Performance of training efficiency among the multi-lengths and the single-length hash codes.

| Method | Average Training Time (s) |
|---|---|
| Single-16 | 155.52 |
| Single-32 | 158.66 |
| Single-48 | 157.27 |
| Single-64 | 158.34 |
| Sum | 629.79 |
| Multiple-[16,32,48,64] | **209.60** |

### 3.5. Comparison to Baselines

We chose nine deep hashing methods to evaluate our method on two datasets, including CNNH [41], DNNH [42], DHN [17], HASHNET [43], DCH [16], MMHH [44], ADSH [45], AMVH [46], and DMLH [31]. By adding a channel attention module and extra experiments, ADMTH may be considered an upgrade to DMLH. Actually, DMLH and ADMTH are multi-task learning methods, and the rest of the methods are single-task. For two different learning methods, we compare their performance experiments on four hash code lengths. The bolded value indicates the optimal performance, whereas the underlined value indicates the suboptimal performance in Tables 4 and 5. The table values are in percentage count units.

The results demonstrate that ADMTH achieves the optimum in most cases. On the one hand, given that the ADMTH better learns the fine-grained information among image categories by integrating the channel attention module, the contained semantic information can be further enriched. On the other hand, the generated multi-length binary codes are optimized by combining the two proposed losses to better mine the semantic relevance involved in these hash codes, which contributes to the promotion of the single hash code characterization capability. Interestingly, we have to admit that the ADMTH generally achieves the optimal performance under the longer length condition. However, it is inferior to a certain method when the number of bits is low. The most likely cause is that the longer length hash code bits can guarantee the model to fully mine and train semantic relationships among hash codes of multiple lengths.

**Table 4.** Retrieval precision compared with baselines presented in MAP on NUS-WIDE.

| Method | NUS-WIDE | | | |
|---|---|---|---|---|
| | **16 Bits** | **32 Bits** | **48 Bits** | **64 Bits** |
| CNNH | 59.0 | 60.0 | 63.5 | 67.0 |
| DNNH | 68.0 | 70.0 | 71.3 | 71.5 |
| DHN | 67.1 | 69.7 | 73.3 | 76.1 |
| HASHNET | 69.5 | 71.5 | 73.8 | 78.0 |
| DCH | 74.0 | 77.2 | 76.9 | 79.3 |
| MMHH | **77.2** | <u>78.4</u> | 78.0 | <u>82.1</u> |
| ADSH | 75.8 | 74.0 | 73.3 | 72.0 |
| AMVH | 72.3 | 74.7 | 75.5 | 77.3 |
| DMLH | 75.0 | 78.2 | <u>79.4</u> | 80.3 |
| ADMTH | <u>76.2</u> | **79.5** | **80.4** | **83.7** |

**Table 5.** Retrieval precision compared with baselines presented in MAP on MS-COCO.

| Method | MS-COCO | | | |
|---|---|---|---|---|
| | 16 Bits | 32 Bits | 48 Bits | 64 Bits |
| CNNH | 56.0 | 56.9 | 53.7 | 50.6 |
| DNNH | 57.7 | 60.2 | 52.3 | 50.1 |
| DHN | 67.5 | 66.8 | 60.0 | 59.8 |
| HASHNET | <u>68.5</u> | 69.0 | 66.4 | 67.8 |
| DCH | **69.6** | **75.7** | <u>72.5</u> | 70.4 |
| ADSH | 57.8 | 63.7 | 60.0 | 56.7 |
| AMVH | 66.7 | 70.0 | 70.3 | <u>72.8</u> |
| DMLH | 65.7 | 70.6 | 70.0 | 72.2 |
| ADMTH | 67.7 | <u>72.8</u> | **73.0** | **74.2** |

*3.6. Parameter Sensitivity*

Figures 5 and 6 illustrate the tendency among precision and parameters $\alpha$, $\beta$ on NUS-WIDE. Within a certain range, the model accuracy is proportional to the parameter value. Once this stability interval is exceeded, the model accuracy is inversely proportional to the parameter value. Eventually, the precision tends to be stable under the suitable range [1,50] to $\alpha$ and range [0.1,10] to $\beta$ ,respectively.

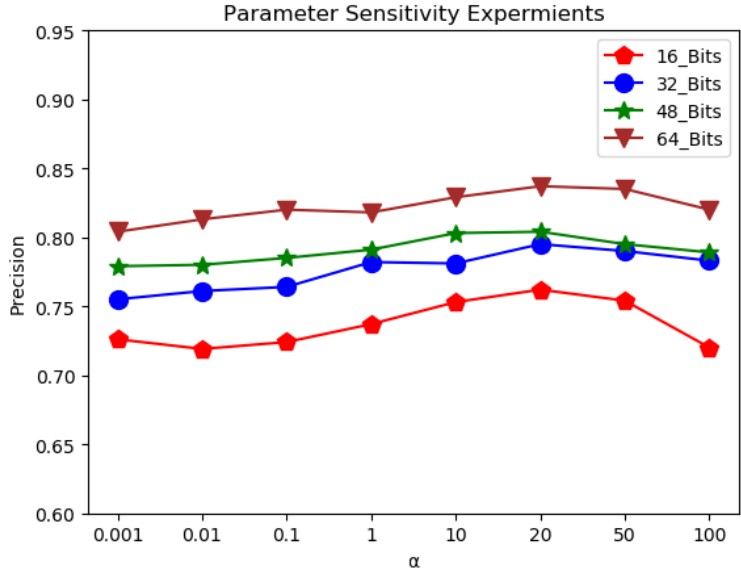

**Figure 5.** Parameter sensitivity curves of ADMTH to $\alpha$.

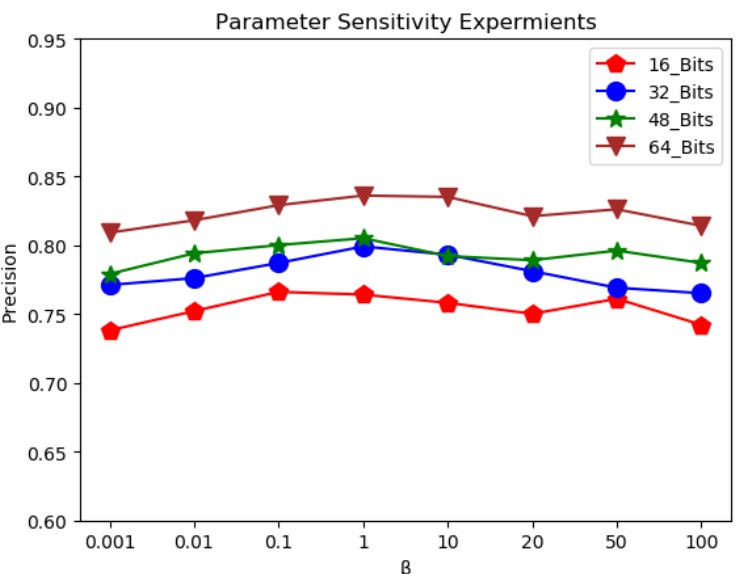

**Figure 6.** Parameter sensitivity curves of ADMTH to $\beta$ .

### 4. Discussion

Generally speaking, by integrating all results of the above experiments, it is novel and effective to cut into hashing from multi-task learning. Firstly, with the help of multiview theory to explore the potential correlation between different hash codes, the design consistency loss narrows the semantic gap within a set of binary codes and fully improves their representation ability. Secondly, a channel attention network is exploited which enhances and renders image features from channel level, making them more discriminative. Both constitute a good baseline for the ADMTH.

From a technical standpoint, the ADMTH needs some improvements to be considered in the future study. Specifically, the consistency loss is designed based on adjacent branches, which describes a linear pairwise similarity relationship. Although this relationship is concise and effective, it may also ignore some nonlinear topological relationships. If we start from the relationship of graph theory, we can use the relationship of forest or graph to describe the information interaction between multi-length hash codes. Secondly, the fusion of CNN and attention mechanism makes up for the lack of learning of local important information by CNN. Nevertheless, due to the increasing number of multimedia data, this fusion way will inevitably lead to an increase in the parameters amounts. Furthermore, we may consider directly patching images and adapting the Transformer backbone architecture. Meanwhile, the learned multi-length hash codes will be directly reflected in the regional semantic information of the image instead of the entire image.

From the overall framework innovation of view, hashing methods can also be integrated with other thinking methods more than just multitask learning. Under the condition of making full use of the low-storage and high-performance characteristics of hashing, absorbing more other mainstream advanced ideas, such as few-shot learning [47,48], reinforcement learning [49,50], transfer learning [51,52], contrastive learning [53,54], etc., so that hashing can be more efficient and adapt quickly downstream tasks, this is undoubtedly a major research direction in the future.

### 5. Conclusions

This paper develops a deep hashing method for image similarity retrieval, namely, ADMTH (attention-oriented deep multi-task hash learning). Specifically, it is directed by the channel attention module and multi-task learning-based hard-parameter-sharing multi-branch network, simultaneously learning multiple lengths hash codes as image representations. Among them, the introduction of channel attention module enriches the fine-grained information in binary codes through implicit operations at the channel level.

The application of multi-task learning makes the model generate multiple hash codes simultaneously and mine associations among these by minimizing consistency loss and pairwise similarity loss. Furthermore, the ADMTH is an end-to-end deep architecture, all modules are embedded in a joint framework, and the parts are interrelated and provide feedback to each other. The ADMTH was experimentally performed on two widely used datasets, demonstrating its efficiency.

Discovered from this study, there is indeed a certain implicit correlation between hash codes of different lengths, and multi-length hashing can be understood as a theoretical application of multi-task learning. Then these hash codes can be regarded as relevant features, which contributes to studying the precise characterization of the hash code from complex semantic scenarios, thereby improving the efficiency and performance of retrieval with hash code.

However, the proposed method still has two restrictions that should be explored gradually in the next work: (1) Enable ADMTH to conduct more detailed study on the relationship among multiple hash codes in a sample, and mine deeper semantic associations. (2) Enable ADMTH to continue to explore the short-length hash code enhancement strategy to promote images' representational ability.

**Author Contributions:** Conceptualization, X.N. and L.W.; coding, L.W. and Z.M.; validation, X.Y. and X.X.; formal analysis, L.W. and F.D.; data curation, Z.M. and L.W.; writing—original draft preparation, L.W.; writing—review and editing, X.N. and L.W. All authors have read and agreed to the published version of the manuscript.

**Funding:** This work is supported in part by the National Natural Science Foundation of China (62176141, 62102235), Shandong Provincial Natural Science Foundation for Distinguished Young Scholars (ZR2021JQ26), Shandong Provincial Natural Science Foundation (ZR2020QF029), Taishan Scholar Project of Shandong Province (tsqn202103088).

**Data Availability Statement:** Not applicable.

**Acknowledgments:** The data presented in this study are available on request from corresponding.

**Conflicts of Interest:** The authors declare no conflict of interest.

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
