# Peer review of "Attention-Oriented Deep Multi-Task Hash Learning"

_electronics, doi:10.3390/electronics12051226_

Round 1
Reviewer 1 Report
This paper seems to be fair only as the plagiarism is 42% on Turnitin.
The manuscript needs to modify and the following observations need to enforce:
1. Problem statement should be included under a separate sub-heading.
2. The discussion section needs to be improved on technical grounds.
The plagiarism rate of this paper is 42% which should be reduced to less than 10%. The report is attached. Find the attachment and reduce the same.

Author Response
Thanks for your comments, the manuscript has been much improved based on these constructive suggestions. Reply comments are attached below in Word version.

Reviewer 2 Report
In this manuscript, the authors proposed attention-oriented deep multi-task hash learning. This article is clear, concise, and suitable for the scope of the journal. Several small suggestions are supplied:
1. Please recheck the manuscript few grammatical mistakes and typographical errors are there.
2. What is the effect of length on hash code? Explain properly
3. Author should explain the basic difference between unsupervised and supervised method.
4. Please revise the conclusion and include what was discovered from the study and how the findings could be applied in the future.
Author Response

(The authors gave the same response as above.)

Reviewer 3 Report
The manuscript proceeds in a straightforward, logical manner, and has been well written. I have only two suggestions.
1. The last sentence in Abstract section has not been clearly written. It would be better if authors could state how the proposed method better than the traditional algorithms.
2. The format of the references should be checked carefully.
Author Response

(The authors gave the same response as above.)

Reviewer 4 Report
Overall, this paper was nicely written and the study was appropriately designed. I don’t have any major concerns regarding your methodology, just a few minor comments that should be addressed to strengthen the paper before it is published. These are summarized below.
-The primary goal of your study is somewhat unclear in the introduction section. The abstract suggests that improving efficiency and reducing runtime are the focus of the paper. However, on Line 37, you suggest accuracy needs to be improved further. Will this be achieved as part of your study or is this simply background information? The context is a little confusing.
-I suggest explaining the novelty of your approach in the introduction section. It is clear your algorithm avoids the need to predefine hash code lengths, but what is unique about your architecture? Are you introducing a new attention mechanism or network structure? You list your contributions on page 2, but not all of these are novel (i.e., some of these steps were implemented in [27-29]), correct?
-There are some minor grammatical errors that should be corrected before publication. For example, the word ‘and’ is missing on Line 46. The first sentence on Line 47 should read ‘Recently’ or ‘In recent years.’ The word ‘designs’ is misspelled on Line 56. The word ‘focus’ should be ‘focuses’ on Line 74. Two spaces are missing on Line 77. The word ‘to’ is missing on Line 132. The word ‘and’ is missing on Line 139. The word ‘want’ should be removed from Line 152. The word ‘the’ is missing on Line 157. A space is missing on Line 168. The word ‘based’ is misspelled on Line 177. Line 196 is missing a period. Line 282 should read ‘learning.’ Line 287 should read ‘the module.’ Line 300 should read ‘both of them.’ Line 309 should read ‘chose.’ Line 312: ‘an upgraded.’ Line 328: ‘the model.’ Section 3.6 should be titled ‘Parameter Sensitivity.’ Lines 345 and 346: missing spaces.
-Your statement on Line 91 would benefit from supporting evidence. What is the underlying theory for this assumption that hash codes can be considered correlated features? Can you elaborate or provide a citation?
-You describe the use of tan() on Line 142, but I believe you intended to use tanh(), as the hyperbolic tangent function is constrained between -1 and 1, while tangent is not. Also, I would label tan() as tanh() on Line 191.
-What is the difference between Eqs. (2) and (3), I didn’t follow how you derived one from the other.
-On Line 230, you describe filtering out images from 21 common categories. Why was this step included? I assume these images were discarded; did you feel they would not provide a valid assessment of your proposed technique?
-On Line 251, you describe MAP as ‘mean retrieval precision,’ but I believe you intended to say ‘mean average precision.’
-Your primary objective for the study is somewhat unclear. As mentioned previously, the abstract focuses on the efficiency of your approach, yet several of your reported results involve either an accuracy metric (i.e., MAP) or a precision measurement (e.g., Figures 5 and 6). I suggest adding a statement near the end of the introduction, explaining your intended application, target data type, and specific goals for your optimized network. What are you trying to achieve and how does it differ from that of previous studies?
Author Response

(The authors gave the same response as above.)

Round 2
Reviewer 1 Report
Observations:
The plagiarism rate of this paper is 20% which should be reduced to less than 10%. Find the attachment and reduce the same.
Rest all corrections included.

Author Response
Thank you for your comments, we have reduced the paper plagiarism rate to 9% and attached a check report. The manuscript has been greatly improved.
